# FlowAR: Scale-wise Autoregressive Image Generation Meets Flow Matching

**Sucheng Ren** [1] **Qihang Yu** [† 2] **Ju He** [† 2] **Xiaohui Shen** [2] **Alan Yuille** [1] **Liang-Chieh Chen** [† 2]

[1]Johns Hopkins University    [2]ByteDance

## Abstract

Autoregressive (AR) modeling has achieved remarkable success in natural language processing by enabling models to generate text with coherence and contextual understanding through next token prediction. Recently, in image generation, VAR proposes scale-wise autoregressive modeling, which extends the next token prediction to the next scale prediction, preserving the 2D structure of images. However, VAR encounters two primary challenges: (1) its complex and rigid scale design limits generalization in next scale prediction, and (2) the generator's dependence on a discrete tokenizer with the same complex scale structure restricts modularity and flexibility in updating the tokenizer. To address these limitations, we introduce FlowAR, a general next scale prediction method featuring a streamlined scale design, where each subsequent scale is simply double the previous one. This eliminates the need for VAR's intricate multi-scale residual tokenizer and enables the use of any off-the-shelf Variational AutoEncoder (VAE). Our simplified design enhances generalization in next scale prediction and facilitates the integration of Flow Matching for high-quality image synthesis. We validate the effectiveness of FlowAR on the challenging ImageNet-256 benchmark, demonstrating superior generation performance compared to previous methods. Codes is available at https://github.com/OliverRensu/FlowAR.

## 1. Introduction

Autoregressive (AR) models have significantly advanced natural language processing (NLP) by modeling the probability distribution of each token given its preceding tokens,

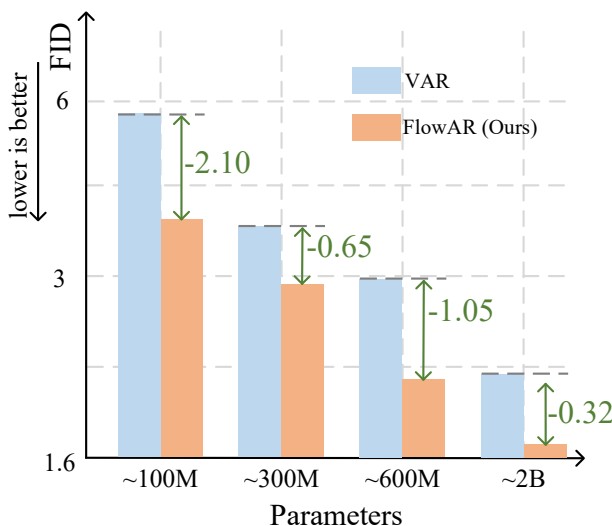

*Figure 1.* **Performance Comparison.** The proposed FlowAR, a general next-scale prediction model enhanced with flow matching, consistently outperforms state-of-the-art VAR (Tian et al., 2024) variants across different model sizes.

allowing for coherent and contextually relevant text generation. Prominent models like GPT-3 (Brown et al., 2020) and its successors (OpenAI, 2022; 2023) have demonstrated remarkable language understanding and generation capabilities, setting new standards across diverse NLP applications.

Building on the success of autoregressive modeling in NLP, this paradigm has been adapted to computer vision, particularly for generating high-fidelity images through sequential content prediction (Esser et al., 2021; Yu et al., 2022; Sun et al., 2024). In these approaches, images are discretely tokenized, with tokens flattened into 1D sequences, enabling autoregressive models to generate images token-by-token. This approach leverages the sequence modeling strengths of AR architectures to capture intricate visual details. However, directly applying 1D token-wise autoregressive methods to images presents notable challenges. Images are inherently two-dimensional (2D), with spatial dependencies across height and width. Flattening image tokens disrupts this 2D structure, potentially compromising spatial coherence and causing artifacts in generated images. Recently, VAR (Tian et al., 2024) addresses these issues by introducing scale-

[1]Johns Hopkins University [2]ByteDance. Correspondence to: Sucheng Ren <sren19@jh.edu>. †: Work done while at ByteDance.

*Proceedings of the 42ⁿᵈ International Conference on Machine Learning*, Vancouver, Canada. PMLR 267, 2025. Copyright 2025 by the author(s).

wise autoregressive modeling, which progressively generates images from coarse to fine scales, preserving the spatial hierarchies and dependencies essential for visual coherence. This scale-wise approach allows autoregressive models to retain the 2D structure during image generation, capturing the layered complexity of visual content more naturally.

Despite its effectiveness, VAR (Tian et al., 2024) faces two significant limitations: (1) a complex and rigid scale design, and (2) a dependency between the generator and a tokenizer that shares this intricate scale structure. Specifically, VAR employs a non-uniform scale sequence, $\{1, 2, 3, 4, 5, 6, 8, 10, 13, 16\}$, where the coarsest scale tokenizes a $256 \times 256$ image into a single $1 \times 1$ token and the finest scale into $16 \times 16$ tokens. This intricate sequence constrains both the tokenizer and generator to operate exclusively at these predefined scales, limiting adaptability to other resolutions or granularities. Consequently, the model struggles to represent or generate features that fall outside this fixed scale sequence. Additionally, the tight coupling between VAR's generator and tokenizer restricts flexibility in independently updating the tokenizer, as both components must adhere to the same scale structure.

To address these limitations, we introduce FlowAR, a flexible and generalized approach to scale-wise autoregressive modeling for image generation, enhanced with flow matching (Lipman et al., 2022). Unlike VAR (Tian et al., 2024), which relies on a complex multi-scale VQGAN *discrete* tokenizer (Razavi et al., 2019; Lee et al., 2022), we utilize any off-the-shelf VAE *continuous* tokenizer (Kingma & Welling, 2014) with a simplified scale design, where each subsequent scale is simply double the previous one (*e.g.*, $\{1, 2, 4, 8, 16\}$), and coarse scale tokens are obtained by directly downsampling the finest scale tokens (*i.e.*, the largest resolution token map). This streamlined design eliminates the need for a specially designed tokenizer and decouples the tokenizer from the generator, allowing greater flexibility to update the tokenizer with any modern VAE (Li et al., 2024; Chen et al., 2024).

To further enhance image quality, we incorporate the flow matching model (Lipman et al., 2022) to learn the probability distribution at each scale. Specifically, given the class token and tokens from previous scales, we use an autoregressive Transformer (Radford et al., 2018) to generate *continuous* semantics that condition the flow matching model, progressively transforming noise into the target latent representation for the current scale. Conditioning is achieved through the proposed *spatially adaptive layer normalization* (Spatial-adaLN), which adaptively adjusts layer normalization (Ba et al., 2016) on a position-by-position basis, capturing fine-grained details and improving the model's ability to generate high-fidelity images. This process is repeated across scales, capturing the hierarchical dependencies inher-

ent in natural images. The final image is then produced by de-tokenizing the predicted latent representation at the finest scale. As shown in Figure 1, Our FlowAR consistently outperform VAR under similar parameters. Specifically, with about 2B parameters our FlowAR outperform VAR by 0.32 FID.

The seamless integration of the scale-wise autoregressive Transformer and scale-wise flow matching model enables FlowAR to capture both the sequential and probabilistic aspects of images with multi-scale information, resulting in improved image synthesis performance. We demonstrate FlowAR's effectiveness on the challenging ImageNet benchmark (Deng et al., 2009), where it achieves state-of-the-art results.

## 2. Related Work

**Autoregressive Models.** Autoregressive modeling (Radford et al., 2018; 2021; Brown et al., 2020; OpenAI, 2023; Touvron et al., 2023a;b; Chowdhery et al., 2023; Team et al., 2023; Ren et al., 2024b;a; 2025b;a; Wang et al., 2024; 2025; Dubey et al., 2024; Bai et al., 2023; Yu et al., 2024a) began in natural language processing, where language Transformers (Vaswani, 2017) are pretrained to predict the next word in a sequence. This concept was first introduced to computer vision by PixelCNN (Van den Oord et al., 2016), which utilized a CNN-based model to predict raw pixel probabilities. With the advent of Transformers, iGPT (Chen et al., 2020) extended this approach by modeling raw pixels using Transformer architectures. VQGAN (Esser et al., 2021; Weber et al., 2024) further advanced the field by applying autoregressive learning within the latent space of VQ-VAE (Van Den Oord et al., 2017), thereby simplifying data representatiown for more efficient modeling (Yu et al., 2024b; Kim et al., 2025). Taking a different direction, Parti (Yu et al., 2022) framed image generation as a sequence-to-sequence task akin to machine translation, using sequences of image tokens as targets instead of text tokens, and leveraging significant advancements in large language models through data and model scaling. LlamaGen (Sun et al., 2024) expanded on this by applying the traditional "next token prediction" paradigm of large language models to visual generation, demonstrating that standard autoregressive models like Llama (Touvron et al., 2023a) can achieve state-of-the-art image generation performance when appropriately scaled, even without specific inductive biases for visual signals. The work most similar to ours is VAR (Tian et al., 2024), which transitioned from token-wise to scale-wise autoregressive modeling by developing a coarse-to-fine next scale prediction. However, VAR faces significant challenges due to its complex scale designs and deep dependency on a scale residual discrete tokenizer. In contrast, our proposed FlowAR employs a simple scale design and maintains compatibility

with any VAE tokenizer (Kingma & Welling, 2014).

**Flow- and Diffusion-based Models.** Diffusion models (Rombach et al., 2022; Peebles & Xie, 2023; Li et al., 2024; Hoogeboom et al., 2023; Ho et al., 2020; Song et al., 2020; Liu et al., 2024; Yang et al., 2024) have surpassed earlier image generation methods like GANs (Goodfellow et al., 2014; Sauer et al., 2022) by utilizing multistep diffusion and denoising processes. Latent Diffusion Models (LDMs) (Rombach et al., 2022) advance this approach by transitioning diffusion from pixel space to latent space, enhancing efficiency and scalability. Building on this foundation, DiT (Peebles & Xie, 2023) and U-ViT (Bao et al., 2023) replace the traditional convolution-based U-Net (Ronneberger et al., 2015) with Transformer architectures within the latent space, further improving performance. Flow matching (Lipman et al., 2022; Liu et al., 2022; Albergo & Vanden-Eijnden, 2022; Shin et al., 2025; He et al., 2025) redefines the forward process as direct paths between the data distribution and a standard normal distribution, offering a more straightforward transition from noise to target data compared to conventional diffusion methods. SiT (Atito et al., 2021) leverages the backbone of DiT and employs flow matching to more directly connect these distributions. Scaling this concept, SD3 (Esser et al., 2024) introduces a novel Transformer-based architecture trained with flow matching for text-to-image generation (Deng et al., 2025). MAR (Li et al., 2024) presents a diffusion-based strategy to model per-token probability distributions in a continuous space, enabling autoregressive models without relying on discrete tokenizers and utilizing a specialized diffusion loss function instead of the traditional categorical cross-entropy loss. In contrast to MAR, our proposed FlowAR employs a scale-wise flow matching model to capture per-scale probabilities, utilizing coarse-to-fine scale-wise conditioning derived from a scale-wise autoregressive model.

**Discussion.** Our FlowAR provides a more flexible framework for next scale prediction, enhanced with flow matching. In Figure 2, we compare our FlowAR model with VAR (Tian et al., 2024), highlighting key differences in both the tokenizer and generator components. For the tokenizer, VAR relies on a multi-scale VQGAN (Razavi et al., 2019; Esser et al., 2021; Lee et al., 2022) that is tightly integrated with its generator and trained on a complex set of scales ($\{1, 2, 3, 4, 5, 6, 8, 10, 13, 16\}$). In contrast, FlowAR can use any off-the-shelf VAE (Kingma & Welling, 2014) as its tokenizer, offering greater flexibility by constructing coarse scale token maps through direct downsampling of the finest scale token map. For the generator, VAR is constrained by a complex and rigid scale structure, whereas FlowAR benefits from a simpler and more general scale design, allowing the integration of the modern flow matching model (Lipman et al., 2022).

# 3. Method

In this section, we begin with an overview of autoregressive modeling in Sec. 3.1, followed by a detailed introduction of the proposed method in Sec. 3.2.

## 3.1. Preliminary: Autoregressive Modeling

**Autoregressive Modeling in NLP.** Consider a corpus represented as a sequence of words $\mathcal{U} = \{w_1, w_2, \ldots, w_n\}$. In NLP, autoregressive models predict each word based on all preceding words in the sequence:

$$p(\mathcal{U}) = \prod_{k=1}^{n} p(w_k \mid w_1, w_2, \ldots, w_{k-1}, \Theta), \qquad (1)$$

where the autoregressive model is parameterized by $\Theta$. The objective function to minimize is the negative log-likelihood over the entire corpus:

$$\mathcal{L} = -\sum_{k=1}^{n} \log p(w_k \mid w_{<k}, \Theta), \qquad (2)$$

where $< k$ denotes all positions preceding $k$. This approach serves as the foundation for successful large-scale language models, as demonstrated in (Touvron et al., 2023a; OpenAI, 2022; 2023).

**Token-wise Autoregressive Modeling for Image Generation.** Extending autoregressive modeling to images involves tokenizing a 2D image $X \in \mathbb{R}^{3 \times H \times W}$ using vector quantization (Van Den Oord et al., 2017). This process converts the image into a grid of discrete tokens, which is subsequently flattened into a one-dimensional sequence $X = \{t_1, t_2, \ldots, t_N\}$:

$$\mathcal{L} = -\sum_{k=1}^{N} \log p(t_k \mid t_{<k}, \Theta). \qquad (3)$$

However, flattening the token grid disrupts the intrinsic two-dimensional spatial structure of the image. To preserve this spatial information, VAR (Tian et al., 2024) introduces a scale-wise autoregressive modeling approach, as described below.

**Scale-wise Autoregressive Modeling for Image Generation.** Rather than flattening images into token sequences, VAR (Tian et al., 2024) decomposes the image into a series of token maps across multiple scales, $S = \{s_1, s_2, \ldots, s_n\}$. Each token map $s_k$ has dimensions $h_k \times w_k$ and is obtained by a specially designed multi-scale VQGAN with residual structure (Razavi et al., 2019; Lee et al., 2022; Tian et al., 2024; Esser et al., 2021). In contrast to single flattened tokens $t_k$ that lose spatial context, each $s_k$ maintains the two-dimensional structure with $h_k \times w_k$ tokens. The autoregressive loss function is then reformulated to predict each

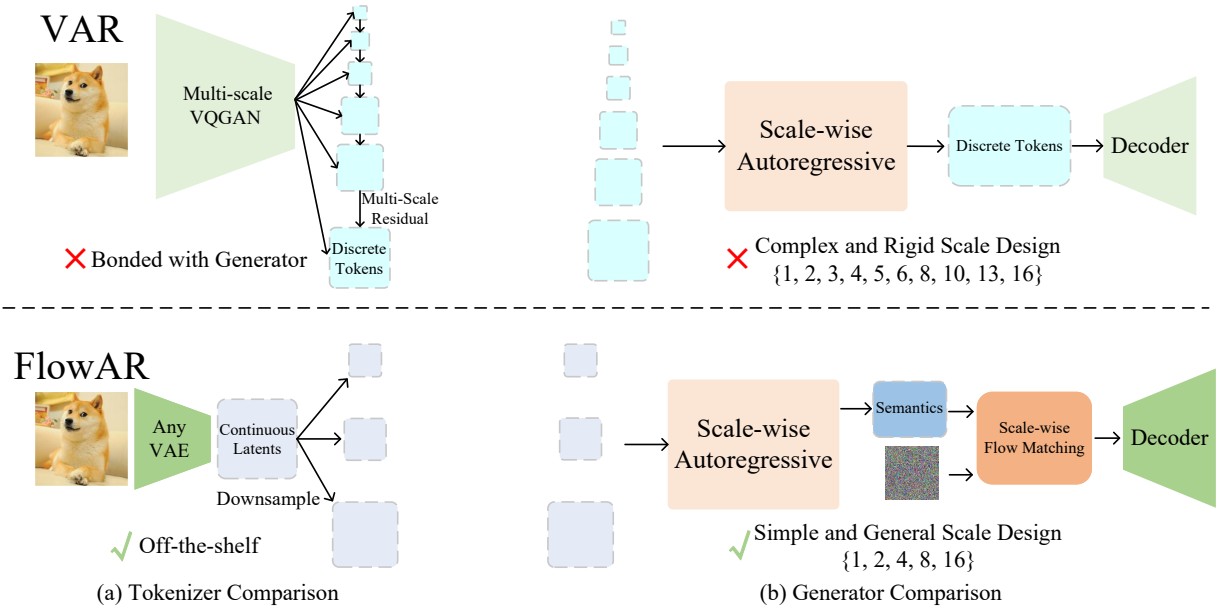

*Figure 2.* **Comparison Between VAR and Our FlowAR** in (a) Tokenizer and (b) Generator design. (a) VAR (Tian et al., 2024) utilizes a complex multi-scale residual VQGAN discrete tokenizer, whereas FlowAR can leverage any off-the-shelf VAE continuous tokenizer, constructing coarse scale token maps by directly downsampling the finest scale token map. (b) VAR's generator is constrained by the same complex and rigid scale design as its tokenizer, while FlowAR benefits from a simple and flexible scale design, enhanced by the flow matching model.

scale based on all preceding scales:

$$\mathcal{L} = -\sum_{k=1}^{n} \log p(s_k \mid s_{<k}, \Theta). \quad (4)$$

In this framework, generating the $k$-th scale in VAR requires attending to all previous scales $s_{<k}$ (*i.e.*, $s_1$ to $s_{k-1}$) and simultaneously predicting all $h_k \times w_k$ tokens in $s_k$ via *categorical* distributions. The chosen scales, $S = \{1, 2, 3, 4, 5, 6, 8, 10, 13, 16\}$, introduce significant complexity to the scale design and constrain the model's generalization capabilities. This is due to the tight coupling between the generator and tokenizer with the scale design, reducing flexibility in updating the tokenizer or supporting alternative scale configurations. Furthermore, VAR's discrete tokenizer relies on a complex multi-scale residual structure, complicating the training process, with essential training code and details remaining publicly unavailable at the time of our submission.

### 3.2. Proposed Method: FlowAR

**Overview.** To address the issues outlined above, we introduce FlowAR seen in Figure 3, a more general scale-wise autoregressive modeling, enhanced with flow matching (Lipman et al., 2022). Our method incorporates two primary improvements over existing next scale prediction (Tian et al., 2024): (1) replacing the multi-scale VQGAN dis-

crete tokenizer with any off-the-shelf VAE continuous tokenizer (Kingma & Welling, 2014), and (2) modeling the per-scale prediction (*i.e.*, predicting all $h_k \times w_k$ tokens in $k$-th scale $s_k$) using flow matching to learn the probability distribution. The first change enables the flexibility to leverage any existing VAE tokenizer, benefiting from recent advances in VAE technology without being constrained by a complex scale sequence design. The second improvement enhances generation quality by utilizing the modern flow matching algorithm.

**Simple Scale Sequence with Any VAE Tokenizer.** Given an image, we extract its *continuous* latent representation $F \in \mathbb{R}^{c \times h \times w}$ using an off-the-shelf Variational Autoencoder (VAE) (Kingma & Welling, 2014). We then construct a set of coarse-to-fine scales by downsampling the latent $F$ as follows:

$$\begin{aligned} S &= \{s^1, s^2, \dots, s^n\} \\ &= \{\text{Down}(F, 2^{n-1}), \text{Down}(F, 2^{n-2}), \dots, \text{Down}(F, 1)\}, \end{aligned} \quad (5)$$

where $\text{Down}(F, r)$ denotes downsampling the latent $F$ by a factor of $r$, and no downsampling is applied when $r = 1$.

Notably, our multi-scale token maps $S$ are derived by directly downsampling the highest-resolution latent $F$, removing the need for complex multi-scale residual VQGAN design (Tian et al., 2024; Razavi et al., 2019; Lee et al.,

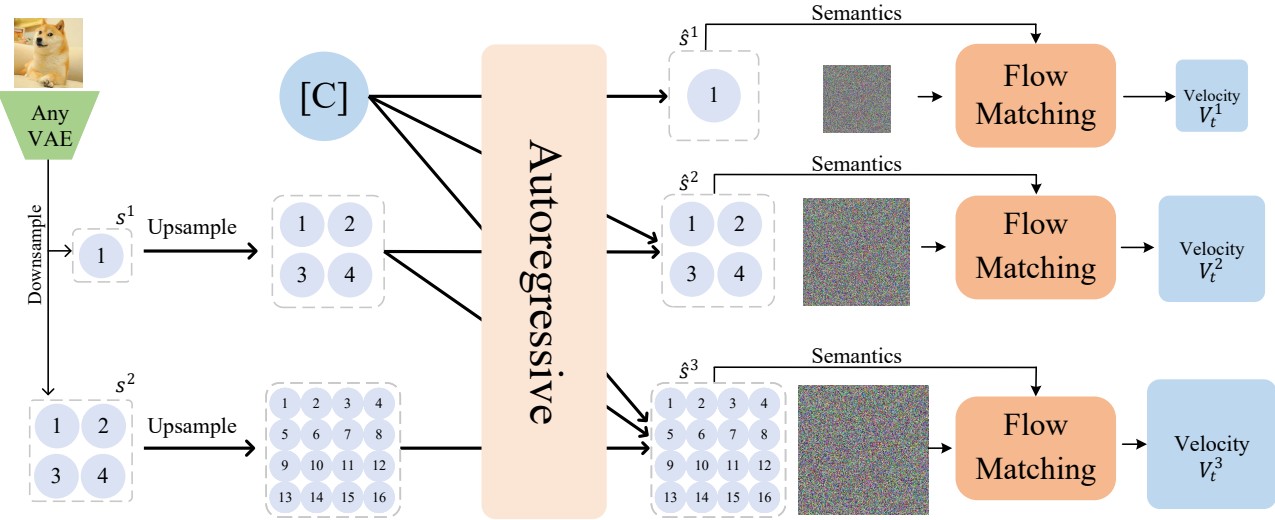

*Figure 3.* **Overview of The Proposed FlowAR.** FlowAR consists of three main components: (1) an off-the-shelf VAE that extracts a continuous latent representation of the image. We then create a set of coarse-to-fine scales by downsampling this latent, forming a sequence of token maps $s^1, s^2, \cdots, s^n$, where each subsequent scale doubles in size from the previous one. (2) A scale-wise autoregressive Transformer that takes as input the sequence $\{[C], \text{Up}(s^1, 2), \ldots, \text{Up}(s^{n-1}, 2)\}$, where $[C]$ is a condition token and $\text{Up}(\cdot, 2)$ denotes upsampling by a factor of 2. This Transformer generates semantic representations for different scales, $\hat{s}^1, \ldots, \hat{s}^n$. (3) A scale-wise flow matching model, conditioned on the semantics $\hat{s}^i$ at each scale $i$ (time step conditions are not shown for simplicity), predicts the velocity given a random time step $t$ that moves the noises to the target data distribution.

2022; Esser et al., 2021). To distinguish our approach, we use the superscript $i$ to denote our $i$-th scale token map, $s^i$. We set $n = 5$ (i.e., $S = \{1, 2, 4, 8, 16\}$), making the scale design in FlowAR significantly simpler and more versatile than that of VAR (Tian et al., 2024). This design allows us to integrate various off-the-shelf VAE tokenizers into our framework, eliminating the need for a multi-scale tokenizer trained on a predefined scale sequence.

With this simplified scale sequence and the flexibility to use any off-the-shelf VAE tokenizer, we introduce our improvement for next scale prediction. Unlike VAR (Tian et al., 2024), which models the *categorical* distribution of each scale using an autoregressive Transformer, we utilize the Transformer to generate *conditioning* information for each scale, while a scale-wise flow matching model captures the scale's probability distribution based on this information. Below, we outline how the autoregressive Transformer generates conditioning information for each scale, followed by details of the scale-wise flow matching module.

**Generating Conditioning Information via Scale-wise Autoregressive Transformer.** To produce the conditioning information for each subsequent scale, we utilize conditions obtained from all previous scales:

$$\hat{s}^i = T\left([C, \text{Up}(s^1, 2), \ldots, \text{Up}(s^{i-1}, 2)]\right), \forall i = 1, \cdots, n, \quad (6)$$

where $T(\cdot)$ represents the autoregressive Transformer model, $C$ is the class condition, and $\text{Up}(s, r)$ denotes the upsam-

pling of latent $s$ by a factor of $r$. For the initial scale ($i = 1$), only the class condition $C$ is used as input. We set $r = 2$, following our simple scale design, where each scale is double the size of the previous one. We refer to the resulting output $\hat{s}^i$ as the *semantics* for the $i$-th scale, which is then used to condition a flow matching module to learn the per-scale probability distribution.

**Scale-wise Flow Matching Model Conditioned by Autoregressive Transformer Output.** Flow matching (Lipman et al., 2022) generates samples from the target data distribution by gradually transforming a source noisy distribution, such as a Gaussian. For each $i$-th scale, FlowAR extends flow matching to generate the scale latent $s^i$, conditioned on the autoregressive Transformer's output $\hat{s}^i$. Specifically, during training, given the scale latent $s^i$ from the target data distribution, we sample a time step $t \in [0, 1]$ and a source noise sample $F_0^i$ from the source noisy distribution, typically setting $F_0^i \sim \mathcal{N}(0, 1)$ to match the shape of conditioned latent $\hat{s}^i$, analogous to the "noise" in diffusion models (Rombach et al., 2022). We then construct the interpolated input $F_t^i$ as:

$$F_t^i = ts^i + (1 - t)F_0^i. \quad (7)$$

The model is trained to predict the velocity $V_t^i$ using $F_t^i$:

$$\begin{aligned} V_t^i &= \frac{dF_t^i}{dt} \\ &= s^i - F_0^i, \end{aligned} \quad (8)$$

where $V_t^i$ indicates the direction to move from $F_t^i$ toward $s^i$,

guiding the transformation from the source to the target distribution at each scale. Unlike prior approaches (Atito et al., 2021; Esser et al., 2024) that condition velocity prediction on class or textual information, we condition on the scale-wise semantics $\hat{s}^i$ from the autoregressive Transformer's output. Notably, in prior methods (Atito et al., 2021; Esser et al., 2024), the conditions and image latents often have different lengths, whereas FlowAR shares the same length (i.e., $s^i$ and $\hat{s}^i$ have the same shape, $\forall i = 1, \cdots, n$). The training objective for scale-wise flow matching is:

$$\mathcal{L} = \sum_{i=1}^{n} \left\| \mathrm{FM} \left( F_t^i, \hat{s}^i, t; \theta \right) - V_t^i \right\|^2, \qquad (9)$$

where FM denotes the flow matching model parameterized by $\theta$. This approach allows the model to capture scale-wise information bi-directionally, enhancing flexibility and efficiency in image generation within our framework.

**Scale-wise Injection of Semantics via Spatial-adaLN.** A key design choice is determining how best to inject the semantic information $\hat{s}^i$, generated by the autoregressive Transformer, into the flow matching module. A straightforward approach would be to concatenate the semantics $\hat{s}^i$ with the flattened input $F_t^i$, similar to in-context conditioning (Bao et al., 2023) where the class condition is concatenated with the input sequence. However, this approach has two main drawbacks: (1) it increases the sequence length input to the flow matching model, raising computational costs, and (2) it provides indirect semantic injection, potentially weakening the effectiveness of semantic guidance. To address these issues, we propose using *spatially adaptive layernorm* for position-by-position semantic injection, resulting in the proposed Spatial-adaLN. Specifically, given the semantics $\hat{s}^i$ from the scale-wise autoregressive Transformer and the intermediate feature $F_t^{i'}$ in the flow matching model, we inject the semantics to the scale $\gamma$, shift $\beta$, and gate $\alpha$ parameters of the adaptive normalization, following the standard adaptive normalization procedure (Ba et al., 2016; Zhang & Sennrich, 2019; Peebles & Xie, 2023):

$$\alpha_1, \alpha_2, \beta_1, \beta_2, \gamma_1, \gamma_2 = \mathrm{MLP}(\hat{s}^i + t),$$
$$\hat{F}_t^{i'} = \mathrm{Attn} \left( \gamma_1 \odot \mathrm{LN}(F_t^{i'}) + \beta_1 \right) \odot \alpha_1,$$
$$F_t^{i''} = \mathrm{MLP} \left( \gamma_2 \odot \mathrm{LN}(\hat{F}_t^{i'}) + \beta_2 \right) \odot \alpha_2, \qquad (10)$$

where $\mathrm{Attn}$ denotes the attention mechanism, and LN denotes the layer norm, $F_t^{i''}$ is the block's output, and $\odot$ is the spatial-wise product. Unlike traditional adaptive normalization, where the scale, shift, and gate parameters lack spatial information, our spatial adaptive normalization provides positional control, enabling dependency on semantics from the

autoregressive Transformer. We apply the Spatial-adaLN to every Transformer block within the flow matching module. Alternative design choices are explored in the ablation study.

**Inference Pipeline.** At the beginning of inference, the autoregressive Transformer generates the initial semantics $\hat{s}^1$ using only the class condition $C$. This semantics $\hat{s}^1$ conditions the flow matching module, which gradually transforms a noise sample into the target distribution for $s^1$. The resulting token map is upsampled by a factor of 2, combined with the class condition, and fed back into the autoregressive Transformer to generate the semantics $\hat{s}^2$, which conditions the flow matching module for the next scale. This process is iterated $n$ scales until the final token map $s^n$ is estimated and subsequently decoded by the VAE decoder to produce the generated image. Notably, we use the KV cache (Pope et al., 2023) in the autoregressive Transformer to efficiently generate each semantics $\hat{s}^i$.

# 4. Experimental Results

In this section, we present our main results on the challenging ImageNet-256 and ImageNet-512 generation benchmark (Deng et al., 2009) (Sec. 4.1), followed by ablation studies (Sec. 4.2).

## 4.1. Main Results

Following the settings in VAR (Tian et al., 2024), we train FlowAR on ImageNet for class-conditional image generation. We evaluate the model using Fréchet Inception Distance (FID) (Heusel et al., 2017) and inception score (IS) (Salimans et al., 2016), Precision (Pre.) (Powers, 2020) and Recall (rec.) (Powers, 2020) as metrics.

**Quantitative Results on ImageNet-256.** As shown in Table 1, when compared to previous generative adversarial models (Brock et al., 2018; Sauer et al., 2022; Kang et al., 2023), autoregressive models (Razavi et al., 2019; Esser et al., 2021; Lee et al., 2022; Sun et al., 2024; Yu et al., 2021), diffusion-based methods (Dhariwal & Nichol, 2021; Rombach et al., 2022; Peebles & Xie, 2023; Bao et al., 2023), and flow matching methods (Atito et al., 2021), FlowAR achieves significant performance gains. Specifically, our best model variant, FlowAR-H, attains an FID of 1.65, outperforming StyleGAN (Sauer et al., 2022) (2.30), LlamaGen-3B (Sun et al., 2024) (2.18), DiT (Peebles & Xie, 2023) (2.27), and SiT (Atito et al., 2021) (2.06).

Compared to the closely related VAR (Tian et al., 2024), FlowAR provides superior image quality at similar model scales. For example, FlowAR-L, with 589M parameters, achieves an FID of 1.90—surpassing both VAR-d20 (FID 2.95) of comparable size and even largest VAR-d30 (FID 1.97), which has 2B parameters. Furthermore, our largest

| method | type | params | FID ↓ | IS ↑ | Pre.↑ | Rec.↑ |
|---|---|---|---|---|---|---|
| BigGAN (Brock et al., 2018) | GAN | 112M | 6.95 | 224.5 | 0.89 | 0.38 |
| GigaGAN (Kang et al., 2023) | GAN | 569M | 3.45 | 225.5 | 0.84 | 0.61 |
| StyleGAN (Sauer et al., 2022) | GAN | 166M | 2.30 | 265.1 | 0.78 | 0.53 |
| ADM (Dhariwal & Nichol, 2021) | diffusion | 554M | 10.94 | 101.0 | 0.69 | 0.63 |
| LDM (Rombach et al., 2022) | diffusion | 400M | 3.60 | 247.7 | - | - |
| U-ViT(Bao et al., 2023) | diffusion | 287M | 3.40 | 219.9 | 0.83 | 0.52 |
| DiT (Peebles & Xie, 2023) | diffusion | 675M | 2.27 | 278.2 | 0.83 | 0.57 |
| SiT (Atito et al., 2021) | flow matching | 675M | 2.06 | 270.3 | 0.82 | 0.59 |
| VQVAE-2 (Razavi et al., 2019) | token-wise autoregressive | 13.5B | 31.11 | 45.0 | 0.36 | 0.57 |
| VQGAN (Esser et al., 2021) | token-wise autoregressive | 1.4B | 15.78 | 74.3 | - | - |
| RQTransformer (Lee et al., 2022) | token-wise autoregressive | 3.8B | 7.55 | 134.0 | - | - |
| LlamaGen-B (Sun et al., 2024) | token-wise autoregressive | 111M | 5.46 | 193.6 | 0.83 | 0.45 |
| ViT-VQGAN (Yu et al., 2021) | token-wise autoregressive | 1.7B | 4.17 | 175.1 | - | - |
| LlamaGen-L (Sun et al., 2024) | token-wise autoregressive | 343M | 3.07 | 256.1 | 0.83 | 0.52 |
| LlamaGen-XL (Sun et al., 2024) | token-wise autoregressive | 775M | 2.62 | 244.1 | 0.80 | 0.57 |
| LlamaGen-3B (Sun et al., 2024) | token-wise autoregressive | 3.1B | 2.18 | 263.3 | 0.81 | 0.58 |
| VAR-d12 (Tian et al., 2024) | scale-wise autoregressive | 132M | 5.81 | 201.3 | 0.81 | 0.45 |
| FlowAR-S | scale-wise autoregressive | 170M | 3.61 | 234.1 | 0.83 | 0.50 |
| VAR-d16 (Tian et al., 2024) | scale-wise autoregressive | 310M | 3.55 | 280.4 | 0.84 | 0.51 |
| FlowAR-B | scale-wise autoregressive | 300M | 2.90 | 272.5 | 0.84 | 0.54 |
| VAR-d20 (Tian et al., 2024) | scale-wise autoregressive | 600M | 2.95 | 302.6 | 0.83 | 0.56 |
| FlowAR-L | scale-wise autoregressive | 589M | 1.90 | 281.4 | 0.83 | 0.57 |
| VAR-d30 (Tian et al., 2024) | scale-wise autoregressive | 2.0B | 1.97 | 323.1 | 0.82 | 0.59 |
| FlowAR-H | scale-wise autoregressive | 1.9B | 1.65 | 296.5 | 0.83 | 0.60 |

*Table 1.* **Generation Results on ImageNet-256.** Metrics reported include Fréchet Inception Distance (FID), inception score (IS), precision (Pre.) and recall (Rec.). The proposed FlowAR demonstrates state-of-the-art generation performance. VAR is evaluated using code and pretrained weights from their official GitHub repository.

model, FlowAR-H (1.9B parameters, FID 1.65), sets a new state-of-the-art benchmark for scale-wise autoregressive image generation.

**Quantitative Results on ImageNet-512.** As demonstrated in Table 2, we present quantitative results on ImageNet-512 generation. Our FlowAR-L model surpasses the closely related VAR-d36 by 0.2 FID while utilizing only 25.6% of its parameters. Compared to other state-of-the-art generative models—including GAN-based BigGAN (Brock et al., 2018), mask-prediction-based MaskGiT (Chang et al., 2022), token-wise autoregressive VQGAN (Esser et al., 2021), and diffusion-based DiT-XL/2 (Peebles & Xie, 2023), our FlowAR-L achieves superior performance across both FID and IS metrics, establishing consistent improvements over all competing approaches.

**Visualization.** We visualize samples generated by FlowAR using different tokenizers in Figure 4, showing that FlowAR is capable of producing high-quality images with impressive visual fidelity and is compatible with various off-the-shelf VAEs. More samples at 256×256 and 512×512 resolution are provided in the Appendix.

| method | params | FID ↓ | IS ↑ |
|---|---|---|---|
| BigGAN (Brock et al., 2018) | 158M | 8.43 | 177.9 |
| MaskGiT (Chang et al., 2022) | 227M | 7.32 | 156.0 |
| VQGAN (Esser et al., 2021) | 227M | 26.52 | 66.8 |
| DiT-XL/2 (Peebles & Xie, 2023) | 675M | 3.04 | 240.8 |
| VAR-d36 (Tian et al., 2024) | 2.3B | 2.63 | 303.2 |
| FlowAR-L | 590M | 2.43 | 295.1 |

*Table 2.* **Generation Results on ImageNet-512.** Metrics reported include Fréchet Inception Distance (FID), inception score (IS). The proposed FlowAR outperforms prior works.

### 4.2. Ablation Studies

We conduct ablation studies on tokenizer compatibility, scale-wise flow matching model and injection of semantics. More ablation studies on construction of scale sequence and scale configurations can be found in the Appendix.

**Tokenizer Compatibility.** VAR (Tian et al., 2024) relies on a complex multi-scale residual tokenizer that compresses images into discrete tokens at different scales, with the scale structure of the tokenizer directly mirroring VAR's architectural scales. This tight coupling between the tokenizer and VAR limits the framework's flexibility and adaptabil-

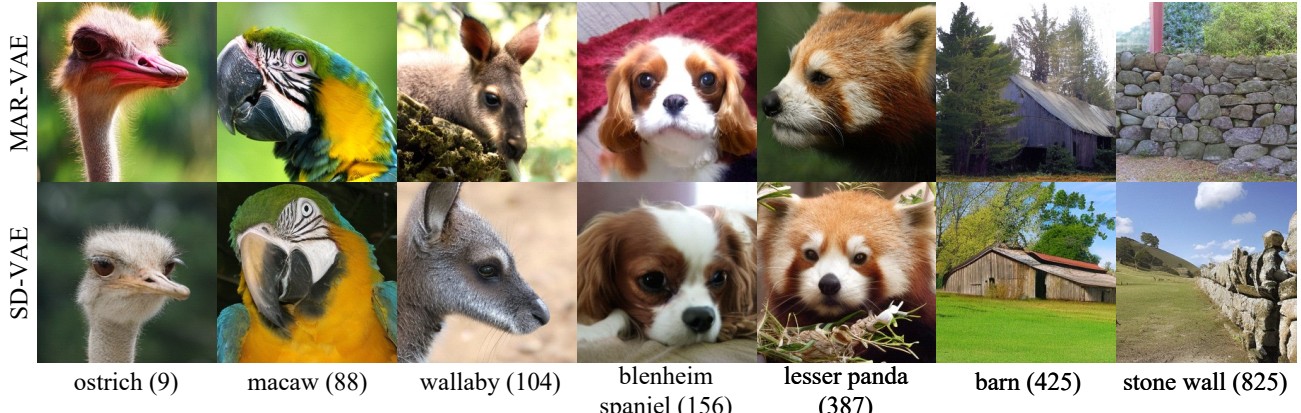

*Figure 4.* **Visualization of Samples Generated by FlowAR Using Different Tokenizers.** FlowAR consistently produces high-quality visual samples across various tokenizer configurations including VAE from MAR (Li et al., 2024) and SD (Rombach et al., 2022).

| tokenizer | Param. | generator | FID | IS |
|---|---|---|---|---|
| multi-scale residual VQGAN | 109.0M | VAR | 5.81 | 201.3 |
| DC-AE (Chen et al., 2024) | 323.4M | FlowAR | 4.22 | 220.9 |
| SD-VAE (Rombach et al., 2022) | 83.7M | FlowAR | 3.94 | 231.0 |
| MAR-VAE (Li et al., 2024) | 66.5M | FlowAR | 3.61 | 234.1 |

*Table 3.* **Ablation on Tokenizer Compatibility.** While VAR (Tian et al., 2024) depends heavily on a multi-scale residual discrete tokenizer, FlowAR is compatible with a variety of continuous VAE tokenizers. The final setting is marked in gray.

| per-token | per-scale | diffusion | flow-matching | FID | IS |
|---|---|---|---|---|---|
| ✓ | | ✓ | | 6.85 | 155.6 |
| ✓ | | | ✓ | 6.15 | 184.1 |
| | ✓ | ✓ | | 3.93 | 223.9 |
| | ✓ | | ✓ | 3.61 | 234.1 |

*Table 4.* **Ablation on Scale-wise Flow Matching Model.** We investigate two design choices in this module: (1) per-token *vs.* per-scale prediction and (2) diffusion *vs.* flow-matching framework. The final setting is highlighted in gray.

ity. In contrast, our proposed FlowAR is compatible with a wide range of variational autoencoders (VAEs), enhancing versatility and ease of integration. As shown in Table 3, FlowAR achieves superior performance across various VAE architectures, including DC-AE (Chen et al., 2024) (FID of 4.22), SD-VAE (Rombach et al., 2022) (FID of 3.94), and MAR-VAE (Li et al., 2024) (FID of 3.61), compared to VAR's multi-scale residual discrete tokenizer, which yields an FID of 5.81. These results underscore FlowAR 's effectiveness and adaptability, highlighting its advantage over VAR's more rigid tokenizer dependency.

**Scale-wise Flow Matching Model.** The flow matching model is used to learn the *per-scale* probability distribution, predicting all $h_k \times w_k$ tokens in the $k$-th scale $s_k$. We consider two design alternatives. First, *per-scale* prediction could be replaced with *per-token* prediction using Multi-Layer Perceptrons (MLPs) (Li et al., 2024), which, however, lacks the ability to capture interactions between tokens. Second, we could substitute the flow matching approach (Lipman et al., 2022) with a diffusion framework (Ho et al., 2020). These design choices are explored in Table 4, where *per-scale* prediction consistently outperforms *per-token* prediction, regardless of whether flow matching or diffusion is used. Additionally, flow matching provides marginal improvements over the diffusion framework. Our final model configuration employs *per-scale* prediction with the flow matching framework.

**Injection of Semantics.** There are several methods to inject the semantics $\hat{s}^i$, generated by the autoregressive Transformer, into the flow matching module. We summarize these methods in Table 5: (1) 'addition': Element-wise addition of the semantics and the flattened input. (2) 'cross attention': Using cross-attention where the flattened input serves as the query and the semantics act as the key and value. (3) 'sequence concatenation': Concatenating the semantics with the flattened input along the sequence dimension. (4) 'channel concatenation': Concatenating the semantics with the flattened input along the channel dimension. (5) 'adaLN': Adaptive LayerNorm conditioned on the spatially averaged semantics. (6) 'Spatial-adaLN': The proposed spatial adaptive LayerNorm, injecting semantics position-by-position.

As shown in Table 5, the choice of semantic injection method significantly impacts performance. The proposed 'Spatial-adaLN' achieves the best results, with an FID of 3.61 and an Inception Score (IS) of 234.1, outperforming all other methods. These results indicate that methods preserving spatial structures and offering position-wise semantic guidance yield superior image generation quality. The exceptional performance of Spatial-adaLN can be attributed to its ability to inject semantics directly into the normalization layers in a spatially adaptive manner.

| injection of semantics | FID | IS |
|---|---|---|
| adaLN | 14.28 | 111.9 |
| sequence concatenation | 9.22 | 146.9 |
| addition | 6.22 | 188.2 |
| cross attention | 5.37 | 202.5 |
| channel concatenation | 4.85 | 210.9 |
| spatial-adaLN | 3.61 | 234.1 |

*Table 5.* **Ablation on Semantic Injection Schemes for the Flow Matching Module.** The proposed Spatial-adaLN injects semantics in a spatially adaptive manner, achieving the best performance. The final setting is highlighted in gray.

## 5. Conclusion

In this work, we presented FlowAR, a flexible and generalized approach to scale-wise autoregressive modeling for image generation, enhanced with flow matching for improved quality. By adopting a streamlined scale design and compatible with any VAE tokenizer, FlowAR addresses limitations of prior models, offering greater adaptability and superior image quality. With spatially adaptive layer normalization, achieving state-of-the-art results on ImageNet-256 generation benchmark. We hope FlowAR will inspire more future research in autoregressive image modeling.

## Impact Statement

FlowAR represents a major advance in scale-wise autoregressive image generation: it eliminates VAR's complex multi-scale residual tokenizer, supports any off-the-shelf Variational AutoEncoder, and adopts a streamlined scale design in which each successive resolution is simply twice that of its predecessor. FlowAR not only elevates the quality of synthetic imagery but also establishes a modular framework that we anticipate will catalyze a new wave of innovation in scalable, adaptable image generation.

### Acknowledge

This work is supported by Office of Naval Research with award N000142412696.

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

# Appendix

The appendix includes the following additional information:

- Sec. A lists the hyper-parameters of FlowAR.

- Sec. B provides the architectural details of FlowAR model variants.

- Sec. C provides more ablation studies.

- Sec. D provides more visualization results.

- Sec. E includes the impact statement.

## A. Hyper-parameters

We list the hyper-parameters of our FlowAR in Table 6.

| training hyper-parameters | |
|---|---|
| optimizer | AdamW |
| warmup epochs | 100 |
| total epochs | 400 |
| batch size | 1024 |
| peak learning rate | 2e-4 |
| minimal learning rate | 1e-5 |
| learning rate schedule | cosine |
| class label dropout rate | 0.1 |
| max gradient norm | 1.0 |

*Table 6.* Hyper-parameters of FlowAR.

## B. Model Variants

In Table 7, we provide four kinds of different configurations of FlowAR for a fair comparison under similar parameters with VAR (Tian et al., 2024). The proposed FlowAR contains two main modules: Autoregressive Model and Flow Matching Model, both build on top of Transformer architectures (Vaswani, 2017).

| variants | autoregressive model | flow matching model |
|---|---|---|
| FlowAR-S | D=12, W=768 | D=2, W= 1024 |
| FlowAR-B | D=16, W= 768 | D=6, W=1024 |
| FlowAR-L | D=16, W= 1024 | D=12, W= 1024 |
| FlowAR-H | D=30, W= 1536 | D=18, W= 1536 |

*Table 7.* **Model Variants.** "D" and "W" represent model depth and width, respectively.

## C. More Ablation Studies

**Construction of Scale Sequence.** Instead of using a multi-scale VQGAN with residual connections (Tian et al., 2024),

| scale construction | FID | IS |
|---|---|---|
| image | 12.19 | 118.2 |
| latent | 3.61 | 234.1 |

*Table 8.* **Ablation on Construction of Scale Sequence.** We compare two approaches for constructing the scale sequence: downsampling the input image before feeding it into the VAE, or downsampling the latents extracted by the VAE. The final setting is highlighted in gray.

| method | scales $S$ | FID | IS |
|---|---|---|---|
| VAR | $\{1, 2, 3, 4, 5, 6, 8, 10, 13, 16\}$ | 5.81 | 201.3 |
| VARs | $\{1, 2, 4, 8, 16\}$ | N/A | N/A |
| FlowAR | $\{1, 2, 4, 8, 16\}$ | 3.61 | 234.1 |
| FlowAR | $\{1, 4, 8, 16\}$ | 4.88 | 200.1 |
| FlowAR | $\{1, 4, 16\}$ | 6.10 | 194.2 |

*Table 9.* **Ablation on Scale Configurations.** We compare VAR and FlowAR under various scale configurations. VAR is constrained by its predefined scale sequence and fails to generalize to other configurations (2nd row), whereas FlowAR demonstrates flexibility across different scale setups. The final setting is highlighted in gray.

as in VAR (Tian et al., 2024), we propose a simpler approach by directly downsampling the latent representations extracted by any off-the-shelf continuous VAE (Kingma & Welling, 2014) tokenizer. An alternative design choice would be to downsample the image before feeding it into a VAE. We explore this design choice in Table 8, where downsampling the latents (FID 3.61) significantly outperforms downsampling the image (FID 12.19).

**Scale Configurations.** To demonstrate FlowAR's flexibility with respect to scale design, we perform an ablation study by progressively reducing the number of scales used in the model. Table 9 presents the results of this study. VAR (Tian et al., 2024) relies on a complex scale configuration with ten scales ($\{1, 2, 3, 4, 5, 6, 8, 10, 13, 16\}$) to achieve its reported performance. Simplifying VAR's scale configuration to $\{1, 2, 4, 8, 16\}$ leads to training failure, indicating a strong dependency between its tokenizer and generator. In contrast, FlowAR demonstrates strong robustness to scale reduction. With the simplified sequence $\{1, 2, 4, 8, 16\}$, FlowAR achieves an FID of 3.61, outperforming VAR even with its full scale sequence. Reducing the scales further to $\{1, 4, 8, 16\}$, FlowAR still maintains competitive performance with an FID of 4.88. Even with just three scales ($\{1, 4, 16\}$), FlowAR achieves an FID of 6.10, comparable to VAR's FID of 5.81.

## D. More Visualization Results

Additional visualization results generated by FlowAR-H are provided from Figure 5 to Figure 10.

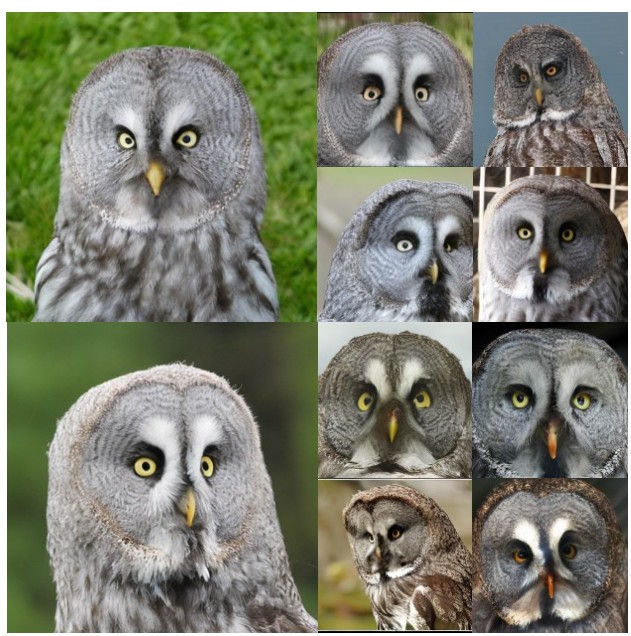

*Figure 5.* **Generated samples from FlowAR.** FlowAR generate high-fidelity great grey owl (24) images.

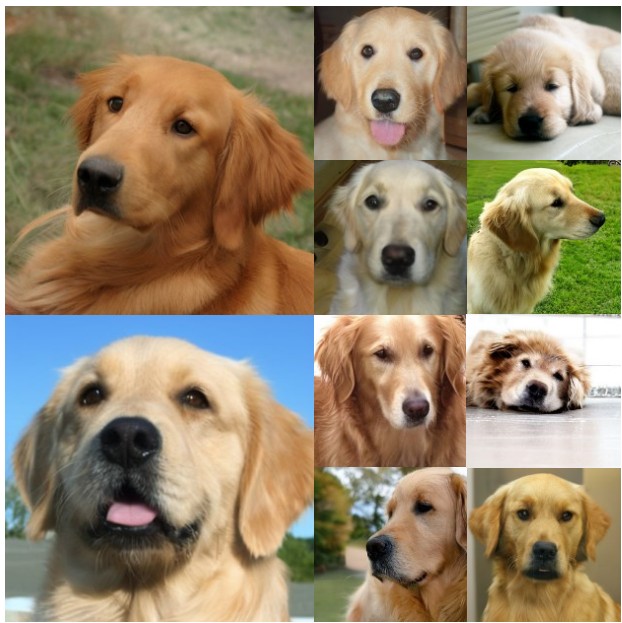

*Figure 7.* **Generated samples from FlowAR.** FlowAR generate high-fidelity golden retriever (207) images.

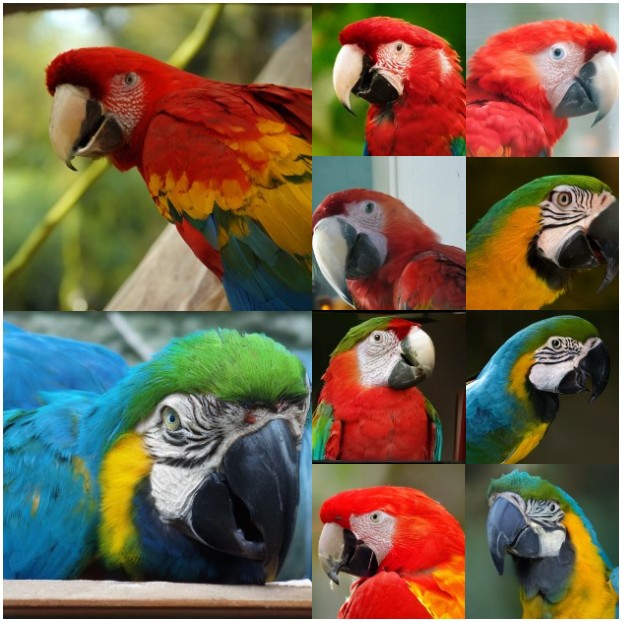

*Figure 6.* **Generated samples from FlowAR.** FlowAR generate high-fidelity macaw (88) images.

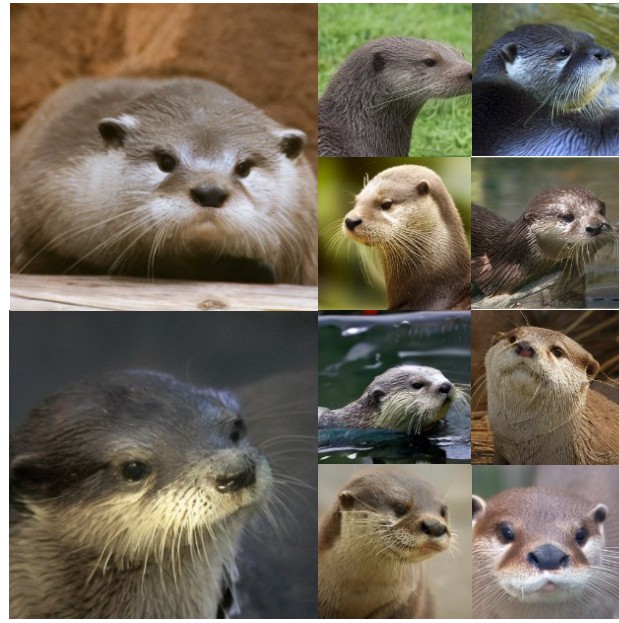

*Figure 8.* **Generated samples from FlowAR.** FlowAR generate high-fidelity otter (360) images.

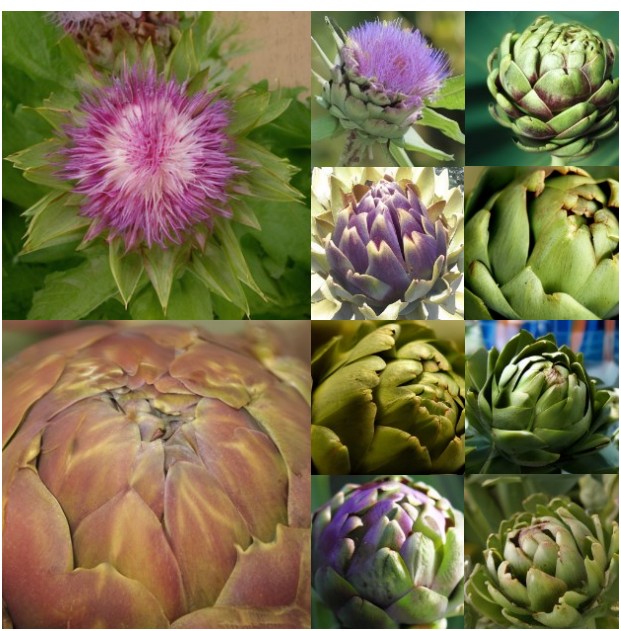

*Figure 9.* **Generated samples from FlowAR.** FlowAR generate high-fidelity artichoke (944) images.

## E. Impact Statement

Our main objective is to advance foundational research on generative models. A direct next step is to extend our method to large-scale visual generation tasks, such as text-to-image or text-to-video models, where it can notably reduce both training and inference costs while enhancing performance. Nonetheless, since our approach derives statistics directly from ImageNet, it may also inherit any biases inherent in that data.

*Figure 10.* **Generated samples from FlowAR.** FlowAR generate high-fidelity alp (970) images.

