# OpenReview forum: "FlowAR: Scale-wise Autoregressive Image Generation Meets Flow Matching"
_ICML.cc/2025/Conference — ICML 2025 poster_

### Official Review · Reviewer_4dzR · 2025-03-13

**Overall Recommendation:** 3

**Summary:**

This paper attempts to address two limitations in VAR work: (1) the complex and rigid scale design and (2) the dependency between the generator and the tokenizer. To address these issues, the paper makes two simplifications: (1) each scale is simply double the previous one; (2) the coarse scale tokens are obtained by directly downsampling the finest scale tokens. These two simplifications make the VAR framework more concise and general, allowing it to be combined with existing continuous tokenizers (VAE) while achieving better generation results despite the simplified design.

**Claims And Evidence:**

Yes

**Essential References Not Discussed:**

The idea of combining flow-matching with multi-scale latents has been widely applied in previous works [1,2]. Additionally, the paradigm of leveraging previous scales to better generate new scales has already been proposed [3]. However, this paper lacks the corresponding citations.

* [1] f-dm: A multi-stage diffusion model via progressive signal transformation, ICLR 2023
* [2] Pyramidal Flow Matching for Efficient Video Generative Modeling, ICLR 2025
* [3] Diffusion forcing: Next-token prediction meets full-sequence diffusion, NeurIPS 2024

**Experimental Designs Or Analyses:**

Yes. I checked the class-conditional image generation results (ImageNet) and ablation studies, including results for 256 and 512 resolutions. These experimental designs and evaluation metrics follow the VAR paper, and I see no issues.

**Methods And Evaluation Criteria:**

Yes. I checked the class-conditional image generation results (ImageNet) and ablation studies, including results for 256 and 512. All benchmarks and metrics follow the VAR paper, and there are no issues with my aspects.

**Other Comments Or Suggestions:**

NA

**Other Strengths And Weaknesses:**

## Strengths:
1. This paper simplifies the complex scale design in VAR, making the VAR framework compatible with any continuous tokenizer (VAE), simplifying the framework to make it more general while also improving the performance.
2. Using autoregression to generate semantics and using them as conditions for diffusion (flow matching) is a reasonable idea.
3. The paper's writing and figures are clear and easy to understand.

## Weaknesses:

1. The authors lack detailed analysis of the proposed scale design. For example, why did the paper simply double the scale rather than triple it? Why were only 5 scales designed? How would the results be affected with more or fewer scales? Providing analysis and results for these design choices would be helpful.
2. The method is essentially a combination of Autoregressive and Diffusion approaches, where Autoregressive integrates information from multiple scales to generate semantics, which then serve as conditions for Diffusion (FlowMatching) generation. Compared to the original VAR, this method might lead to slower generation speed. However, the paper lacks experiments comparing generation speeds and related analysis.
3. In Table 1, FlowAR's performance advantage mainly comes from FID and IS, but the improvements in Precision and Recall metrics are not substantial. Could the authors provide more analysis and explanation for this experimental phenomenon?
4. The direct comparison of different tokenizers in Table 3 may not be entirely fair, as different tokenizers have different parameter counts. It would be helpful if the authors added parameter count comparisons for the Tokenizers (VAEs).
5. The "Any VAE" in Figure 2 might be ambiguous; it's suggested that the authors change it to "Any continuous tokenizer (VAE)".

**Questions For Authors:**

NA

**Relation To Broader Scientific Literature:**

NA

**Theoretical Claims:**

There are no theoretical claims in this paper.

---

> ### Author Rebuttal · Authors · 2025-04-01
>
> We thank the reviewer for the constructive feedback and address the concerns below.
>
> > Essential References
>
> Thank you for the suggestions. FlowAR focuses on ***next-scale*** prediction through ***autoregressive modeling***, whereas the referenced works explore diffusion-based approaches with multi-stage architectures [1], multi-scale designs [2], or independent noise levels combined with causal next-token prediction [3]. We will cite and discuss all related works accordingly.
>
> > W1: scale design analysis
>
> We have ablated the scale design in Table 9 of the Appendix (page 11 in paper). For quick reference, we provide the results below. As shown, even with just three scales ({1, 4, 16}), FlowAR achieves an FID of 6.10—comparable to VAR, which uses a much more complex scale configuration ({1, 2, 3, 4, 5, 6, 8, 10, 13, 16}) and obtains an FID of 5.81.
>
> |method |scales | FID |
> |:--:|:--:|:--:|
> | VAR| {1, 2, 3, 4, 5, 6, 8, 10, 13, 16} |5.81 |
> | VAR| {1, 2, 4, 8, 16}| N/A |
> | FlowAR| {1, 2, 4, 8, 16} |3.61|
> | FlowAR| {1, 4, 8, 16} |4.88 |
> | FlowAR| {1, 4, 16} |6.10|
>
> > W2: speed comparison with VAR
>
> In the table below, we compare the generation speed and performance of FlowAR with VAR and other widely used diffusion- and flow-matching-based models, including MAR, DiT, and SiT.
>
> |model|params|inference time (sec/image)|FID|
> |:---:|:---:|:---:|:---:|
> |DiT-XL|675M|1.7|2.26|
> |SiT-XL|675M|1.7|2.06|
> |MAR-B|208M|1.25|2.31|
> |VAR-d30|2B|0.07|1.97|
> |FlowAR-L|589M|0.12|1.90|
> |FlowAR-H|1.9B|0.24|1.65|
>
> As shown, although FlowAR involves additional denoising steps due to flow matching and is slightly slower than VAR, FlowAR-L remains over 10× faster than other diffusion- and flow-matching-based models such as MAR, DiT, and SiT while also achieving superior image generation quality.
>
> > W3: Precision and Recall metrics in Table 1
>
> Precision and Recall metrics often saturate and offer limited additional insight beyond FID. Therefore, following prior works such as MAR, VAR, DiT, and SiT, we primarily focus on optimizing and reporting FID, which remains the most sensitive and informative metric for assessing image generation quality in our evaluations.
>
> > W4: Update Table 3 with tokenizer’s parameters
>
> Thank you for the suggestion. In the updated table below, we include the tokenizer parameters to provide a more complete and transparent comparison.
>
> |tokenizer|tokenizer params |generator |FID|
> |:---:|:---:|:---:|:---:|
> | multi-scale residual VQGAN|109.0M |VAR |5.81 |
> |DC-AE | 323.4M | FlowAR|4.22 |
> |SD-VAE |83.7M | FlowAR|3.94 |
> | MAR-VAE|66.5M | FlowAR|3.61 |
>
> > W5: Change “Any VAE” to “Any continuous tokenizer (VAE)”
>
> Thank you for the suggestion. We will update the text to “any continuous tokenizer (VAE)” accordingly.

---

> > ### Comment · Reviewer_4dzR · 2025-04-03
> >
> > Thanks for the rebuttals. My questions are well answered. I would keep my original rating.

---

### Official Review · Reviewer_6Cuw · 2025-03-13

**Overall Recommendation:** 3

**Summary:**

In this work, the authors propose FlowAR which generates images sequentially at different scales. FlowAR first generates conditional vector of different resolution using AR model and then relies on flow matching model to generate clean image of corresponding resolution. Unlike VAR, FlowAR is more flexible which doesn't rely on curated VAE to acquire the latent space. On standard ImageNet generation benchmark, FlowAR achieves competitive performance comparing to SoTA baselines.

**Claims And Evidence:**

Yes, claims are supported by convincing evidence.

**Essential References Not Discussed:**

N/A

**Experimental Designs Or Analyses:**

Yes, experimental designs are sound.

**Methods And Evaluation Criteria:**

Yes, the proposed methods are properly evaluated.

**Other Comments Or Suggestions:**

See "Questions For Authors".

**Other Strengths And Weaknesses:**

See "Questions For Authors".

**Questions For Authors:**

1. What is the inference speed of FlowAR compared with baselines like VAR. How much overhead is there from the flow-matching module?
2. Can the authors give more detailed description about the AR and flow-matching Transformer? For example, is it based on the implementation of VAR?
3. Did the authors have ablation study on how to balance sizes of AR and flow-matching models? How are their sizes determined?
4. Are there visualization of samples generated of different scales in FlowAR which could help better demonstrate what FlowAR learns to generate images.
5. In table 3, FlowAR with MAR-VAE achieves the best performance. However, MAR used VAR from LDM which is less powerful than SD-VAE. Does authors have any assumptions about why this is the case?

**Relation To Broader Scientific Literature:**

The paper is related to diffusion/flow matching model as well as autoregressive model for image generation.

**Theoretical Claims:**

N/A

---

> ### Author Rebuttal · Authors · 2025-04-01
>
> We thank the reviewer for the constructive feedback and address the concerns below.
>
> > Q1: speed comparison with VAR
>
> While FlowAR requires additional denoising steps due to the use of flow matching, it is only slightly slower than VAR, and FlowAR-L still achieves a significant 10× speedup over other diffusion- and flow-matching-based models such as MAR, DiT, and SiT.
>
> |model|params|inference time (sec/image)|FID|
> |:---:|:---:|:---:|:---:|
> |DiT-XL|675M|1.7|2.26|
> |SiT-XL|675M|1.7|2.06|
> |MAR-B|208M|1.25|2.31|
> |VAR-d30|2B|0.07|1.97|
> |FlowAR-L|589M|0.12|1.90|
> |FlowAR-H|1.9B|0.24|1.65|
>
> > Q2: more detailed description about the AR and flow-matching Transformer
>
> Our autoregressive and flow matching modules are built upon the VAR and SiT codebases, respectively. As stated in the paper, we will ***fully open-source*** the training and inference code, along with model checkpoints, to enable the community to reproduce our results and examine all implementation details.
>
> > Q3: balance sizes of AR and flow-matching
>
> Thank you for the question. We ablate the effect of AR and flow matching model sizes in the table below. Increasing the size of the flow matching module initially improves performance, but further scaling eventually leads to a performance drop.
>
> |model|AR params|Flow matching params|inference time (sec/image)|FID|
> |:---:|:---:|:---:|:---:|:---:|
> |FlowAR-L|504M|70M|0.03 |2.32|
> |FlowAR-L|411M|143M| 0.07|2.01|
> |FlowAR-L(default setting)|309M|280M| 0.12 |1.90|
> |FlowAR-L|152M|420M| 0.19|1.98|
>
> > Q4: visualization of samples
>
> While our primary focus is on the image quality at the final scale, we provide intermediate-scale visualizations at [anonymous link](https://anonymous.4open.science/r/Visualization-1A5C/README.md) and will add more intermediate-scale visualizations in the final version.
>
> > Q5: VAE from MAR
>
> To analyze the impact of different underlying VAEs, we report their reconstruction FID (rFID) in the table below:
>
> |tokenizer|rFID|
> |:--:|:--:|
> |MAR-VAE|0.53|
> |SD-VAE|0.82|
>
> As shown, MAR-VAE achieves a better rFID, indicating better reconstruction quality, which correlates with improved generation FID (gFID) in both our experiments and those reported in the MAR paper.
>
> To further validate the effect of MAR-VAE versus SD-VAE, we also compare FlowAR with another generator, UViT, using both tokenizers. MAR-VAE consistently delivers better generation performance across different generator architectures.
>
> |model|MAR-VAE|SD-VAE|gFID|
> |:--:|:--:|:--:|:--:|
> |UViT|√||3.24|
> |UViT||√|3.52|
> |FlowAR-S|√||3.61|
> |FlowAR-S||√|3.94|

---

### Official Review · Reviewer_U6ur · 2025-03-13

**Overall Recommendation:** 4

**Summary:**

This paper proposes a multi-scale approach for image generation by combining  autoregressive modeling with flow matching at each scale. Instead of using a VQVAE-based tokenization as in the VAR approach, the method uses continuous latents from a VAE which are downsampled to get tokens at different scales. An autoregressive transformer takes as input the condition and multiscale representations encoding semantics at different scale. Conditioned on this, the velocity vectors are predicted. Experiments are performed on the ImageNet dataset where the approach outperforms prior work.

**Claims And Evidence:**

+The work claims to address the problem of multiscale autoregressive image generation. To this end, limitation of prior work are identified and a new framework is proposed for autoregressive image generation
+ The method section is clearly written and provides substantial argument in support of the design choices, for example, VAE encoder for multiscale semantics instead of a VQ-VAE based approach.
+Experimental results on the ImageNet dataset show the effectiveness of the approach.

**Essential References Not Discussed:**

Literature on multiscale image generation is not new and the work should discuss them in the related work.
For example.
[a] PixelCNN models with auxiliary variables for natural image modeling. ICML 2017

[b] Generating high fidelity images with subscale pixel networks and multidimensional upscaling. ICLR, 2019.

[c] MaCow: Masked convolutional generative flow. NeurIPS, 2019

[d] Pixelpyramids: Exact inference models from lossless image pyramids. ICCV 2021

**Experimental Designs Or Analyses:**

+The experimental setup is inline with prior work and compared wrt to the FID and IS scores.
+Ablations are performed to validate the design choices.

**Methods And Evaluation Criteria:**

+To address the problem of image generation within the framework of autoregressive modeling the idea of using VAE over VQVAE with streamlined upsampling procedure compared to prior work is good.
+The evaluation is based on comparison of the visual fidelity of the generated images and the parameter overhead with respect to the baseline.
+ Adequate ablations are performed.

**Other Comments Or Suggestions:**

No

**Other Strengths And Weaknesses:**

+The work is incremental but provides a good solution to advance the field of autoregressive image generation using transformers.
+The paper is very well written and follows the limitations of prior work which are well addressed with new modeling and formulation.
- Related work does not do justice to the prior work on multiscale generation even though prior work are not based on transformers, they are still relevant and should be discussed.

**Questions For Authors:**

1. How does the method scale with resolution? Are there any limitations on the resolutions eg, square images in the VAE based framework considered?
2. Are same number of timesteps used for flow-matching across image scales?
3. How well does the approach work for long-text conditioning?
4. How are the positional encodings handled with the VAE based approach and transformers?
5. What is the level of controllability that can be handled with these models at different scales? for complex images and conditions how does the model ensure faithful generation across scales?

**Relation To Broader Scientific Literature:**

The work is related to  multiscale autoregressive image generation within the transformer backbone. The motivation is the success of transformers and autoregressive modeling in NLP. This has gained traction for modeling images.

**Theoretical Claims:**

The work does not make any new theoretical claims. The efficiency of continuous representations from VAE compared to the code-book look-up of the VQ-VAE is well-established.

---

> ### Author Rebuttal · Authors · 2025-04-01
>
> We thank the reviewer for the constructive feedback and address the concerns below.
>
> > Essential References
>
> We thank the reviewer for the suggestion. These works represent early efforts in applying autoregressive modeling to image generation in pixel space. We will cite and discuss them in the related work section.
>
> > Q1: scale with resolution
>
> In our design, each subsequent scale is simply double the previous one—providing a simple and intuitive hierarchical structure. Additionally, our method supports non-square resolutions, though this may result in a slight drop in performance.
>
> |scales|FID|
> |:---|:---:|
> |$\{1\times 1, 2\times 2, 4\times 4, 8\times 8, 16\times 16\}$ (default setting)|1.90|
> |$\{1\times 1, 2\times 4, 4\times 8, 8\times 16, 16\times 16\}$|2.21|
> |$\{1\times 1, 4\times 2, 8\times 4, 16\times 8, 16\times 16\}$|2.16|
>
> > Q2: time steps across scales
>
> Thank you for the question. By default, we use ***the same number of time steps*** (25 steps) for all scales. We ablate the effect of using different denoising time steps across scales (1x1, 2x2, 4x4, 8x8, 16x16) in the following table.
>
>
> |steps at 1X1|steps at 2X2|steps at 4X4|steps at 8X8|steps at 16X16|inference time (sec/image)|FID|
> |:---:|:---:|:---:|:---:|:---:|:---:|:---:|
> |25|25|25|25|25|0.12|1.90|
> |50|50|50|50|50|0.24|1.89|
> |15|15|15|15|15|0.08|2.08|
> |10|10|10|10|10|0.06|2.48|
> |15|15|20|25|25| 0.12|1.96|
> |25|25|20|15|15| 0.08|1.94|
>
> As shown, increasing the time steps from 25 to 50 across all scales results in only a marginal FID improvement of 0.01, while doubling the inference time. Reducing the steps to 15 or 10 significantly speeds up inference but comes at the cost of degraded image quality. We also evaluate varying the number of steps across scales. Using fewer steps for smaller scales leads to a slight FID drop of 0.06 without affecting inference speed. Conversely, assigning fewer steps to larger scales also slightly degrades performance but yields faster inference. Overall, these results suggest that a uniform 25-step setting offers a better trade-off between generation quality and efficiency.
>
> > Q3: long-text conditioning
>
> Since FlowAR is an autoregressive model, we can either concatenate the long text condition in front of the image sequence—similar to other autoregressive models [A]—or incorporate cross-attention as done in diffusion models [B].
>
> [A] Autoregressive Model Beats Diffusion: Llama for Scalable Image Generation
>
>
> [B] High-Resolution Image Synthesis with Latent Diffusion Models
>
> > Q4: positional encodings
>
> The positional embeddings are handled in the same way as prior methods [C, D, E, F] (e.g., absolute position embeddings are added at the input layer). As stated in the paper, the detailed implementation (for both training and inference) will be fully open-source, allowing the community to reproduce and examine all details.
>
> [C] Visual Autoregressive Modeling: Scalable Image Generation via Next-Scale Prediction
>
> [D] Autoregressive Image Generation without Vector Quantization
>
> [E] Alleviating Distortion in Image Generation via Multi-Resolution Diffusion Models and Time-Dependent Layer Normalization
>
> [F] Scalable Diffusion Models with Transformers
>
> > Q5: level of controllability
>
> We thank the reviewer for the insightful question. In this work, our goal is to develop a general and broadly applicable framework by treating all scales uniformly—for example, using the same number of denoising steps across scales—on the standard ImageNet benchmark, following the next-scale prediction framework introduced by VAR. While datasets such as LAION or COCO may benefit from dataset-specific scale designs, our focus is on establishing a simple and general scale design that can generalize across settings without requiring dataset-specific tuning.

---

### Official Review · Reviewer_Ry1o · 2025-03-13

**Overall Recommendation:** 3

**Summary:**

The paper explores image generation using a next-scale prediction objective, in which image latents (from a VAE) are progressively super-resolved from 1x1 scale to 2x2, 4x4, and so on. To that end, the authors present a method in which an autoregressive model predicts a conditioning for a per-scale flow-matching model. Unlike models like VAR that rely on a specifically trained residual multi-scale tokenizer, FlowAR works on down-sampled latents from arbitrary VAEs, which makes it broadly applicable. On class-conditional ImageNet generation, FlowAR outperforms VAR and other baselines, and achieves SotA performance for next-scale prediction models.

## update after rebuttal
The authors addressed most of my concerns and I raised my rating accordingly.

**Claims And Evidence:**

Overall the claims are sensible, yet, there are a few (FLOPS) controlled baselines that would be important to ablate to properly verify the contribution of the method:
1. What's the importance of the autoregressive model in the method? How would a baseline perform in which there is no AR model, but all scales can attend to each other and all scales are diffused at once like Matryoshka Diffusion Models [Gu et al., 2023]? Of course, in this case the input is not the fully predicted previous scales, but the noised version of all the scales at once.
2. Similar to before, how then would a FLOPS-controlled single-scale flow model perform? Is there a need for multi-scale prediction?
3. I appreciate the per-token prediction ablation in L413, but how would the performance change if the semantics s were not predicted in a next-scale manner, but fully autoregressive? I.e. autoregressive on a inter-scale, and on a intra-scale level (raster-scan).
4. Like pyramidal flow [Jin et al., 2024], what if there were no split between AR and flow models, but instead you predict the flow between upsampled previous scale and next scale directly?
5. What is the performance and runtime trade-off when trading-off AR model and flow model layers? MAR shows that the diffusion layer can be rather small, but in FlowAR they are somewhat large.

**Essential References Not Discussed:**

There are a few important missing references. "Next-scale" training and prediction has been a long-standing research area, going back to at least early explorations with GANs [1,2]. It has also been studied from the perspective of cascaded super-resolution with diffusion models, e.g. [3,4], as well as quite recently and relevant to FlowAR in videos with flow models [5]. VAR has also had relevant follow-up work, e.g. [6]. More related to the autoregression + continuous targets direction, [7,8] are also relevant works in that direction, with [7] building upon the cited MAR work. I would also note [9] as a quite relevant work that performs diffusion on multiple scales at once, instead of in an autoregressive manner.

[1] Deep Generative Image Models using a Laplacian Pyramid of Adversarial Networks, Denton et al., 2015

[2] Progressive Growing of GANs for Improved Quality, Stability, and Variation, Karras et al., 2017

[3] Cascaded Diffusion Models for High Fidelity Image Generation, Ho et al., 2021

[4] Photorealistic Text-to-Image Diffusion Models with Deep Language Understanding, Saharia et al., 2022

[5] Pyramidal Flow Matching for Efficient Video Generative Modeling, Jin et al., 2024

[6] Infinity: Scaling Bitwise AutoRegressive Modeling for High-Resolution Image Synthesis, Han et al. 2024

[7] Fluid: Scaling Autoregressive Text-to-image Generative Models with Continuous Tokens, Fan et al., 2024

[8] DART: Denoising Autoregressive Transformer for Scalable Text-to-Image Generation, Gu et al., 2024

[9] Matryoshka Diffusion Models, Gu et al., 2023

**Experimental Designs Or Analyses:**

The paper gives enough implementation details to make a reasonable attempt at reimplementing it, but not enough to be fully confident of the exact training procedure and settings. It is also not very detailed in the ablation settings.

**Methods And Evaluation Criteria:**

The submission demonstrates the proposed method in a class-conditional generation framework with ImageNet-1k. This setting is very commonly used, and under that umbrella the evals make sense. That said, class-conditional ImageNet generation is a benchmark that is somewhat over-optimized, and the common generation metrics (gFID against train-set statistics) favor and measure overfitting to the train set. Indeed, FlowAR surpasses the train-val FID of 1.78, as reported by VAR. At that point, I would argue that the benchmark becomes nearly meaningless, and more scalable and less data-starved benchmarks are required. Commonly, fully autoregressive models can quickly overfit with just a few passes over the same data, and the FlowAR trains for 400 epochs. This might differ with hypbrid AR+Flow models, but it is also not clear whether any data augmentation was used.
I would also like to note here that it is unclear to me if the Fig 1 results should be compared like that. As far as I can tell, VAR was trained for a shorter number of epochs. How did the authors ensure these two settings are comparable?

**Other Comments Or Suggestions:**

Generally the manuscript is well written and easy to read. I did not spot many obvious typos.
I would suggest to show more visuals, and if possible, show visuals of intermediate stage reconstructions (even if they may not be fully valid).

**Other Strengths And Weaknesses:**

The paper presents strong results on IN1K class-conditional generation. The method is overall straight-forward, and greatly simplifies some of the limitations of VAR's need for a specially trained multi-scale tokenizer. Indeed, it can be run with a wide variety of standard VAE models. That said, I feel that there are a few important baselines missing, and I have doubts about the validity of the IN1K class-conditional generation benchmark when pushing models to such high performance levels. Train-set gFID to me is not a solid metric, and it looks like VAR significantly out-performs FlowAR in terms of IS. The setting and the rank-reversal between metrics makes me have some doubts about the more fine-grained side of results, but overall the method is conceptually simple and seems to work reasonably well.

**Questions For Authors:**

On L288: "it provides indirect semantic injection, potentially weakening the effectiveness of semantic guidance." This claim is not very clear to me. Could the authors expand upon the reasoning for this? In which sense is the "semantic injection" indirect?

**Relation To Broader Scientific Literature:**

This submission is timely, given the community's recent interest into alternative generation schemes (other than diffusion and raster-scan next-token prediction) that explore hierarchical, or multi-scale approaches. The work is very related to VAR and classical "progressive super-resolution" works (e.g. cascaded diffusion, Imagen), as well as recent literature that explores next-token prediction with continuous latents (e.g. GIVT, MAR, Fluid).

**Theoretical Claims:**

The submission does not make any theoretical claims. All claims are empirical in nature.

---

> ### Author Rebuttal · Authors · 2025-04-01
>
> We thank the reviewer for the constructive feedback.
>
> > C1: importance of AR
>
> Unlike Matryoshka Diffusion Models (MDM), which use a nestedUNet to jointly denoise all scales, FlowAR conditions on previously generated scales. Below, we include a baseline, “diffuse all noisy scales” (without AR), which faces two issues: (1) ***slower inference***, as denoising is applied to all scales, resulting in extremely long sequences; and (2) ***worse FID***, due to the absence of guidance from previously generated scales.
>
> |model|params|inference time (sec/image)| FID|
> |:---:|:---:|:---:|:---:|
> |MDM|434M|3.16|3.51|
> |diffuse all noisy scale|600M|2.80|1.98|
> |FlowAR-L|589M|0.12|1.90|
>
> > C2: single-scale flow model
>
> Below, we compare with SiT-XL, a pure single-scale flow model, and “FlowAR-L w/ single-scale.” Our multi-scale design outperforms SiT-XL while requiring 14% of the Flops and achieving a 13.9× speedup. Slightly slower than the single-scale variant, the multi-scale design achieves a 1.78 FID improvement.
>
> |model| params | FLOPS |inference time (sec/image)|FID |
> |:---:|:---:|:----:|:----:|:----:|
> |SiT-XL| 675M |58.3T |1.67|2.06|
> |FlowAR-L w/ single-scale| 589M |  6.2T| 0.09  |3.68  |
> |FlowAR-L w/ multi-scale  | 589M |8.2T  |0.12|1.90|
>
> FLOPs are computed across the entire generation process.
>
> > C3: token-wise prediction of semantics
>
> This token-wise design results in slower inference and worse performance:
>
> |model | inference time (sec/image)| FID|
> |:---:|:---:|:----:|
> |token-wise AR | 0.58 |3.02|
> |scale-wise AR (ours) | 0.12|1.90|
>
> > C4: no split between AR and flow models
>
> Below, we experimented with directly predicting the flow between the upsampled previous scale and the next scale.
>
> |model| inference time (sec/image)|FID |
> |:---:|:---:|:----:|
> |no split  | 0.45|2.45 |
> |with split (ours)  |0.12|1.90|
>
> > C5: performance vs. runtime
>
> Unlike MAR, which models each token individually, FlowAR models the probability over entire scales. It also reduces inference time by using only 5 AR steps (with 25 denoising steps each), compared to MAR’s 256 AR steps (with 100 steps each). As shown below, FlowAR-H is 5× faster than MAR-B while achieving a 0.66 FID improvement.
>
> |model|params|inference time(sec/image) |FID|
> |:---:|:---:|:----:|:----:|
> |MAR-B|208M|1.25|2.31|
> |FlowAR-H|1.9B|0.24|1.65|
>
> Due to response length limitation, please also refer to our response to Reviewer 6Cuw’s Q3 for the ablation on AR and flow model sizes.
>
> > ImageNet train-set gFID
>
> We thank the reviewer for raising this point. FID is a widely used metric in prior works (e.g., MAR, DiT, VAR, SiT, StyleGAN), and our setup follows these standards for fair comparison.
>
> > Training settings of VAR and FlowAR
>
> For ***data augmentation***, we follow VAR’s setup with RandomCrop and horizontal flipping. As noted in the paper, we will ***fully open-source*** our code and checkpoints for reproducibility and transparency.
> Below, we clarify the training settings of VAR and FlowAR. While diffusion and flow models typically require long training (e.g., MAR: 800 epochs, DiT/SiT: 1400), FlowAR achieves strong performance with just 400 epochs. Compared to VAR-d30/d24 (350 epochs), FlowAR-H trains for a similar duration but uses nearly half the GPU hours, thanks to its simpler scale design and shorter sequences. The table below summarizes A100 training hours and FID scores. Compared to VAR-d20/12, our 250-epoch version of FlowAR achieves better performance while incurring only half the training cost.
> |model|epochs | params|training hours| FID|
> |:--:|:--:|:--:|:--:|:--:|
> |VAR-d30| 350 | 2B|9657 hours| 1.97 |
> |FlowAR-H| 350| 1.9B| 4667| 1.70|
> |FlowAR-H| 400 | 1.9B|5334 hours| 1.65 |
> |VAR-d12|250|132M  |788 hours  | 5.81|
> |FlowAR-S|250|170M|401 hours |4.12|
> |FlowAR-S|400|170M|642 hours| 3.61|
> |VAR-d20|250|600M  |3012 hours |2.95|
> |FlowAR-L|250|589M  | 1460 hours | 2.15 |
> |FlowAR-L|400|589M  |2335 hours  |1.90 |
>
> > Essential References
>
> We will cite and discuss all mentioned works in the related work section and compare with [3, 8, 9] in the experiments. Our work is most closely related to VAR and its follow-up [6], which autoregressively predict the next scale—unlike other methods that, while leveraging multi-scale information, do not adopt this formulation.
>
> > IS vs. FID
>
> We observe an IS–FID trade-off when tuning the CFG scale. While we focus on optimizing FID, a slight adjustment still allows both FID and IS to outperform VAR by a notable margin.
>
> |model|cfg|FID|IS|
> |:---:|:---:|:----:|:----:|
> |VAR|2.4|1.97| 323.1
> |FlowAR-H|2.4|1.65| 296.5|
> |FlowAR-H|3.0|1.75|357.3|
>
> > Visualization
>
> Due to response length limitation, please also refer to our response to Reviewer 6Cuw’s Q4 for the visualization of intermediate stage generated samples.
>
>
> > L288 claim
>
> Our Spatial-adaLN injects scale- and position-wise semantics for fine-grained conditioning, unlike the simpler concat approach (L281), which provides only sequence-level, indirect guidance.

---

> > ### Comment · Reviewer_Ry1o · 2025-04-03
> >
> > I thank the authors for their detailed rebuttal and clarifying my questions. Most of my concerns are addressed, except for C5. My question there was more about the trade-off between AR model size and flow model size. For example, in Appendix B the authors list FlowAR-H's model size as being split into a 30-layer AR model and 18 layer flow model. What's the impact of assigning more or less capacity to the AR / flow model?

---

> > > ### Author Response · Authors · 2025-04-03
> > >
> > > As mentioned in previous Rebuttal, due to response length limitation, we ablate the size of AR and Flow matching model sizes in the response to ***Reviewer 6Cuw’s Q3***. For your convenience, we provide the table below.
> > >
> > > As shown, increasing the size of the flow matching module initially improves performance, but further scaling eventually leads to a performance drop.
> > >
> > > |model|AR params|Flow matching params|inference time (sec/image)|FID|
> > > |:---:|:---:|:---:|:---:|:---:|
> > > |FlowAR-L|504M|70M|0.03 |2.32|
> > > |FlowAR-L|411M|143M| 0.07|2.01|
> > > |FlowAR-L (default setting)|309M|280M| 0.12 |1.90|
> > > |FlowAR-L|152M|420M| 0.19|1.98|
> > > |FlowAR-H| 1620M| 321M|  0.13| 1.74|
> > > |FlowAR-H (default setting)| 1280M| 633M| 0.24 |1.65|
> > > |FlowAR-H| 855M| 1048M| 0.35 | 1.67|

---

### Decision · Program_Chairs · 2025-05-01

**Decision:**

Accept (poster)

**Comment:**

The paper proposes FlowAR, a multi-scale image generation approach combining autoregressive modeling with flow-matching, showing strong performance improvements on class-conditional ImageNet generation compared to previous methods such as VAR. Reviewers generally agree that the paper makes valuable contributions. Given the positive reviewer consensus post-rebuttal, I recommend acceptance for this paper.